# In Vitro Antifungal Activity of LL-37 Analogue Peptides against *Candida* spp.

**DOI:** 10.3390/jof8111173

**Published:** 2022-11-07

**Authors:** Gladys Pinilla, Yenifer Tatiana Coronado, Gabriel Chaves, Liliana Muñoz, Jeannette Navarrete, Luz Mary Salazar, Carlos Pelleschi Taborda, Julián E. Muñoz

**Affiliations:** 1Grupo de Investigación REMA, Faculty of Health Sciences, Universidad Colegio Mayor de Cundinamarca, Calle 28 # 5-B-02, Bogotá 110311, Colombia; 2Departamento de Química, Facultad de Ciencias, Universidad Nacional de Colombia, Carrera 30 # 45-03, Ciudad Universitaria, Bogotá 111321, Colombia; 3Department of Microbiology, Biomedical Sciences Institute, University of São Paulo, São Paulo 05508-000, Brazil; 4Laboratory of Medical Mycology, Institute of Tropical Medicine of São Paulo LIM53/Medical School, University of São Paulo, São Paulo 4023-062, Brazil; 5Studies in Translational Microbiology and Emerging Diseases (MICROS) Research Group, Translational Medicine Institute, School of Medicine and Health Sciences, Universidad del Rosario, Calle 63 C # 26-63, Bogotá 111221, Colombia

**Keywords:** antimicrobial peptides, LL-37, analogue peptides, antifungal therapy, candidiasis, *Candida* spp.

## Abstract

Fungal infections have increased in recent decades with considerable morbidity and mortality, mainly in immunosuppressed or admitted-to-the-ICU patients. The fungal resistance to conventional antifungal treatments has become a public health problem, especially with *Candida* that presents resistance to several antifungals. Therefore, generating new alternatives of antifungal therapy is fundamental. One of these possibilities is the use of antimicrobial peptides, such as LL-37, which acts on the disruption of the microorganism membrane and promotes immunomodulatory effects in the host. In this study, we evaluated the in vitro antifungal activity of the LL-37 analogue peptides (AC-1, LL37-1, AC-2, and D) against different *Candida* spp. and clinical isolates obtained from patients with vulvovaginal candidiasis. Our results suggest that the peptides with the best ranges of MICs were LL37-1 and AC-2 (0.07 µM) against the strains studied. This inhibitory effect was confirmed by analyzing the yeast growth curves that evidenced a significant decrease in the fungal growth after exposure to LL-37 peptides. By the XTT technique we observed a significant reduction in the biofilm formation process when compared to yeasts untreated with the analogue peptides. In conclusion, we suggest that LL-37 analogue peptides may play an important antimicrobial role against *Candida* spp.

## 1. Introduction

Candidiasis is one of the most medically important mycoses worldwide with different clinical manifestations; one of the most frequent is the invasive candidiasis, responsible for about 75% of opportunistic yeast infections in hospitalized patients with inherent risk factors [1]. Although there are about 200 species of *Candida* described, only some can cause infection and, in some cases, present reduced susceptibility to antifungals as well as a recognized intrahospital environment infectivity. The main etiological agent of invasive candidiasis is *C. albicans*, representing about 50% of cases. However, the prevalence of *non-albicans Candida* species such as *C. glabrata* in the United States and northwest Europe and *C. parapsilosis* in Latin America, southern Europe, India, and Pakistan is increasing [2].

Candidiasis is a common cause of morbidity and mortality. In the United States, nosocomial infection by *Candida* spp. is the fourth most common cause of hospital admission [2]. In Colombia (2017), there were 753,000 cases of fungal infections, of which about 600,000 cases were candidiasis, 130,000 were aspergillosis, and 16,000 cases were of opportunistic infections associated to patients infected with HIV [3].

Epidemiological data obtained in Latin America indicate that the incidence rates of candidiasis are higher than those described in North America and Europe. A study carried out in 11 Brazilian hospitals showed a candidemia global incidence of 2.49 cases per 1000 hospital admissions [4]. Another study performed in 22 hospitals in different Latin American countries showed an incidence of 0.98 cases per 1000 hospital admissions with a wide variation among countries (from 0.32 in Chile to 1.75 in Argentina) [5].

The global increase in fungal infections leads to a proportional increase in resistance to antifungal drugs such as those of the azole family, frequently used in clinics. Troublingly, resistance to these drugs has been detected in different clinically relevant *Candida* species [4], generating the requirement to investigate and implement new antifungal therapy alternatives to mitigate this problem. Recently, the use of antimicrobial peptides from different organisms such as insects, amphibians, mammals, plants, and even the same host (human) with fungicidal effect has been proposed, specifically against yeasts of the genus *Candida* with clinical importance [5]. Glands, structural cells, and the immune system have the ability to produce and secrete some molecules that have antimicrobial properties, such as ß-defensins, histatins, and cathelicidins [6].

The LL-37 peptide is a cationic human cathelicidin with antimicrobial properties. Its chemical structure begins with two leucines and is composed of 37 amino acids (LLGDFFRKSKEKIGKEFKRIVQRIKDFLRNLVPRTES) [7]. Its α-helical cationic amphipathic structure gives it an antimicrobial effect that acts on the microorganism’s cellular structures [8]. The LL-37 peptide has exhibited a broad spectrum of antimicrobial activity against *S. aureus*, *M. tuberculosis*, and *L. monocytogenes* [7] among other Gram-positive and Gram-negative bacteria [9], viral agents such as herpes simplex virus type 1 (HSV-1) and adenovirus [10], fungi such as *C. albicans* [11], and some parasites such as *Leishmania* sp. [12]. The antimicrobial profile of the LL-37 peptide is mediated in two ways: transmembrane pore formation and intracellular damage. Additionally, LL-37 has other activities related to host defense, including inflammatory response regulation, adaptive immune cells’ chemotactic migration to the infection site, endotoxin neutralization, angiogenesis, and phagocytic cell activation in the presence of different reactive oxygen species (ROS), thus producing the cell death [10,11,13].

The LL-37 peptide originates from a precursor, an 18 kD molecule called hCAP-18, which exists constitutively in the neutrophil granules [14]. LL-37 can also be secreted by epithelial cells present in the skin, gastrointestinal tract, and respiratory tract. Additionally, it can be produced by natural killer cells [7] as well as by subpopulations of monocytes and lymphocytes [15]. The LL-37 peptide is an important component in the epidermis and can act on keratinocytes to induce the pro-inflammatory cytokines’ and chemokines’ production [16].

Recently, a group of short peptides derived from LL-37 composed of 12, 20, and 30 amino acids was designed and showed significant antimicrobial activity, such as that observed in an experimental model against *Entamoeba hystolitica*, in which these short peptides were used in a range of concentration between 10 and 50 µM to control the parasite [17]. Additionally, other studies by our group showed promising antimicrobial effects of the LL-37 analogue peptides that were tested in medically important bacterial strains such as *Staphylococcus* spp. and *Pseudomonas aeruginosa*.

An important characteristic of these short peptides derived from LL-37 is that they have a smaller hemolytic effect than its natural structure [17]. The analogue peptides of LL-37 (LL37-1, AC-1, AC-2, and D) have a shorter peptide structure, which confers them protection against proteases secreted by some microorganisms as well as those secreted by the host. The LL37-1 peptide is 24 amino acids long and amidated at the carboxyl-terminal position. The ACL 37-1 (AC-1) peptide is 23 amino acids long and presents acetylation at the amino terminal portion and amidation at the carboxyl-terminal domain. The ACL 37-2 (AC-2) peptide, 24 amino acids long, begins with glycine and presents acetylation at the amino terminal domain and amidation at the carboxyl-terminal position. Finally, the DL 37-2 (D) peptide with 25 amino acids has a modification in the amino terminal position that turns this analogue peptide into a D enantiomer.

It is important to note that the positive charge, peptide structural variations (for example, D enantiomer), and the short peptide chains could decrease the active sites where the endoproteases (released by the microorganisms as a defense mechanism) give the analogue peptides an intrinsic resistance to microorganisms [18]. Therefore, the LL-37 analogue peptides mentioned were selected in this study in order to evaluate their possible antifungal effect against different reference species of *Candida* as well as 20 strains with clinical importance that cause vulvovaginal candidiasis.

## 2. Materials and Methods

### 2.1. Synthesis and Purification of Antimicrobial Peptides

The human cathelicidin LL-37-derived peptides (AC-1, AC-2, LL37-1, and D) with the amidated C-terminal portion were obtained in Peptide 2.0 (Chantilly, VA, USA). Analyses with a high-performance liquid chromatography system (HPLC) and mass spectrometry (MS) performed by the manufacturer showed that the analogue synthetic human cathelicidin was 98% pure. Peptides derived from the LL-37 bioinformatic design were investigated on an anti-BP server (http://www.imtech.res.in/raghava/antibp/index.htmL) (accessed on 20 January 2019) using the APD database (http://aps.unmc.edu/AP/main.php) (accessed on 20 January 2019). To carry out the in silico experiments, we analyzed 15 different fragments and then selected the four peptides with the best antimicrobial profile: LL37-1 (GRKSAKKIGKRAKRIVQRIKDFLR) and AC-2 (GRKSAKKIGKRAKRIVQRIKDFLR), both with 24 amino acids, with the difference being that the AC-2 peptide presented acetylation in the terminal amino group; AC-1 (RKSKEKIGKEFKRIVQRIKDFLR) with 23 amino acids; and D ((d-PHE) GRKSAKKIGKRAKRIVQRIKD (d-F) LR) with 25 amino acids.

### 2.2. Microorganisms

Standard strains of *C. albicans* (SC5314 and ATCC 10231), *C. parapsilosis* ATCC 22019, *C. krusei* ATCC 6558, and *C. tropicalis* ATCC 750 were challenged. An azole-resistant strain of *C. albicans* 256 was identified by MALDI-TOF and then incorporated into the study. Additionally, 20 clinical isolates of *C. albicans* from patients with vulvovaginal candidiasis from Bogotá, Colombia, were analyzed. All strains were preserved in 10% glycerol at −80 °C. Three days before the experiments started, each fungal strain was sub-cultured in Sabouraud dextrose agar (Becton, Dickinson and Company; Sparks, NV, USA) and maintained at 37 °C for 24–48 h. Subsequently, isolated colonies of each strain were sub-cultured in brain–heart infusion liquid medium (BHI, Becton Dickinson, New Jersey, NJ, USA) and shaken at 100 rpm for 24 h at 37 °C in order to recover exponentially growing yeasts.

### 2.3. Susceptibility Assay of Candida Planktonic Cells

The minimum inhibitory concentration (MIC) was determined by the liquid medium microdilution technique, described in the M27-S4 document (Clinical and Laboratory Standards Institute (CLSI), 2012). The MIC was defined as the lowest necessary concentration of the LL-37 analogue peptides (AC-1, AC-2, LL37-1, and D) capable of inhibiting the different Candida species’ growth (*C. albicans* SC5314 and ATCC 10231, *C. parapsilosis* ATCC 22019, *C. krusei* ATCC 6558, and *C. tropicalis* ATCC 750) and the clinical strains evaluated in this study. Yeasts were re-suspended in RPMI 1640 medium (BiowHITTAKER^®^, Lonza, Belgium) supplemented with (3-(n-morpholino) propanesulfonic acid (MOPs, Sigma-Aldrich, Missouri, MO, USA) at a 0.5 McFarland scale representing 1 × 10^8^ colony-forming units per milliliter (CFU/mL). Antimicrobial-derived peptides of LL-37 were added at 100 µM, 50 µM, 25 µM, 12.5 µM, 6.25 µM, 3.12 µM, and 1.5 µM concentrations, diluted in RPMI medium, and placed into sterile 96-well microtiter polystyrene plates (Corning Incorporated, New York, NY, USA) until adjusting the volume to 100 µL. Fluconazole (FLZ) (Pfizer, New York, NY, USA) at an initial concentration of 64 µg/mL and amphotericin B (AMB) (Sigma-Aldrich, Missouri, MO, USA) at an initial concentration of 16 µg/mL were used as antifungal controls since they are frequently used in the treatment of different mycoses including candidiasis. As growth control, we utilized different *Candida* yeasts without any antifungal treatment or the LL-37 analogue antimicrobial peptides. After 48 h of samples’ incubation, the 492 nm optical density measurement was carried out using a FC multiscan spectrophotometer (Thermo Fisher Scientific Inc., Waltham, MA, USA).

### 2.4. Determination of Growth Phases Using the LL-37-Derived Peptides

Growth curves were performed using different *Candida* species: *C. albicans* ATCC 10231, *C. albicans* SC5314, *C. krusei* ATCC 6558, and *C. parapsilosis* ATCC 22019. These yeasts were re-suspended in RPMI 1640 medium supplemented with MOPs in a scale of 0.5 McFarland or its equivalent in optical density from 0.08 to 0.1, which represents 1 × 10^8^ CFU/mL, utilizing an FC multiscan spectrophotometer (Thermo Fisher Scientific Inc., Waltham, MA, USA) to measure the samples at a wavelength of 600 nm. Subsequently, in sterile 100-well microtiter polystyrene plates (Honeycomb, Thermo Fisher Scientific, Inc., Waltham, MA, USA), 150 μL of MOPs-supplemented RPMI medium and 150 μL of each yeast inoculum were added separately to determine the yeasts’ growth in the presence of the LL-37-derived peptides (AC-1, LL37-1, AC-2, and D) at concentrations of 10 μM, 5 μM, 2.5 μM, 1.25 μM, and 0.62 μM. For this experiment, the amphotericin B (Sigma-Aldrich, USA) was used as antifungal control in an initial concentration of 40 μg/mL. Samples were incubated and analyzed in a BioScreen C piece of equipment (Thermo Labsystems Type FP-1100-C, Waltham, MA, USA) at a constant temperature of 37 °C with continuous shaking for 48 h. All measurements were carried out in an automated way every hour through the presence of turbidity at a wavelength of 600 nm. To increase the reproducibility, each assay parameter was performed in triplicate.

### 2.5. Preformed Biofilm Eradication Assay

The yeast concentration of *C. albicans* 10231 was adjusted to 1 × 10^6^ cells/mL in RPMI medium supplemented with MOPs. Subsequently, 100 μL/well of this solution was placed in 96-well plates and incubated for 24 h at 37 °C to induce the biofilm formation. Afterward, 100 µL of AC-1, LL37-1, AC-2, and D analogue peptides were added at different concentrations (20 μM, 10 μM, 5 μM, 2.5 μM, 1.25 μM, 0.62 μM, 0.31 μM, 0.15 μM), respectively, as well as the antifungal control (amphotericin B) in concentrations of 64 to 0.5 µg/mL, which is equivalent to 0.069 to 0.0005 ≈ µM. After this process, samples were incubated again for 24 h at 37 °C. Finally, the analysis of the peptides anti-biofilm effect was carried out by means of the reduction assay with 2,3-bis (2-methoxy-4-nitro-5sulfo-phenyl)-2H-tetrazolium-5-carboxanilide (XTT, Sigma-Aldrich, Missouri, MO, USA) in incubation for 3 h and subsequent OD measurement at 620 nm as described by Pierce and coworkers (2008) [19].

### 2.6. Scanning Electron Microscopy (SEM)

*C. albicans* ATCC 10231 was treated with a concentration lower than the MIC of each antimicrobial peptide, analogous to LL-37 for 24 h at 37 °C. Afterward, the samples were fixed in 2.5% glutaraldehyde for 3 h at room temperature. The samples were then applied on a polylysine-coated coverslip, serially dehydrated in alcohol, and subsequently observed in a scanning electron microscope and focused ion beam FE-MEB LYRA3 of TESCAN (Brno, Czech Republic), which has an integrated X-ray energy dispersive spectroscopy (EDS) microanalysis system (energy dispersive X-ray spectroscopy).

### 2.7. Statistical Analysis

Statistical analysis was performed using GraphPad Prism version 7.05 (GraphPad Software, San Diego, CA, USA). Statistical comparisons were carried out by the analysis of variance (one-way ANOVA) followed by a Tukey–Kramer post hoc test. The *p*-values of <0.05 indicated statistical significance.

## 3. Results

### 3.1. Antifungal Susceptibility in Planktonic Cells of Candida spp.

The susceptibility of *C. albicans* ATCC 10231 and SC5314, *C. parapsilosis* ATCC 22019, *C. krusei* ATCC 6558, *C. tropicalis* ATCC 750, and the clinical isolates herein studied, which were exposed to different concentrations of the LL-37 analogue peptides, can be observed in Table 1. The AC-1 peptide demonstrated antifungal activity due to the high susceptibility of most of the strains challenged. Species such as *C. parapsilosis*, *C. krusei*, and *C. tropicalis* presented an MIC of 0.15 µM. However, the clinical isolates and the *C. albicans* SC5314 strain presented a lower susceptibility from the AC-1 peptide with an MIC up to 10 µM. Regarding the AC-2 and LL37-1 peptides, a high susceptibility of the *C. albicans* strains with MICs from 0.07 to 5 µM was evidenced; only a few clinical strains had a higher range (10 µM). Finally, a promising result was obtained from the D peptide against *C. tropicalis* and *C. albicans* ATCC 10231 with MIC values equivalent to 0.15 and 1.25 µM, respectively. The least promising effect of the D peptide was against *C. krusei* ATCC 6558 with an MIC of 10 µM. On the other hand, fluconazole, used as a control, showed inhibition in most of the reference strains with the exception of *C. krusei* due to its intrinsic resistance to this antifungal. Some clinical strains also showed low sensitivity to fluconazole. However, we highlight the significant antifungal effect of LL-37 analogue peptides against clinical isolates from patients with vulvovaginal candidiasis that were less sensitive to the antifungal drug used as control. Our results showed that all the strains included in this study were susceptible to amphotericin B, with the highest MIC being 4 µg/mL in the case of *C. albicans* 256 and some clinical isolates, as can be observed in Table 1.

### 3.2. Determination of Yeast Growth Phases Using LL-37 Analogue Peptides

*Candida* spp. were exposed to different concentrations of AC-1, AC-2, LL37-1, and D antimicrobial peptides, aiming to analyze the fungal growth curves. In all experiments, fluconazole was used as antifungal control. Our results showed that the yeast susceptibility profiles were variable, as observed in the previous section. The growth curves of the *C. albicans* ATCC 10231 strain demonstrated a promising antifungal effect by the four LL-37-derived peptides, with a significant decrease in growth at concentrations of 10, 5, 2.5, 1.25, and 0.62 µM (Figure 1).

In the case of *C. albicans* 256 and the clinical isolate with a high azoles resistance profile, the AC-1 and LL37-1 peptides showed the most efficient antimicrobial activity at all concentrations used. Although the enantiomer D did not show an excellent performance, it was able to reduce the yeast growth at 2.5, 5, and 10 µM. On the other hand, the AC-2-derived peptide did not present any growth inhibitory effect on *C. albicans* 256 (Figure 2).

When the peptides derived from LL-37 were tested against the *C. albicans* SC5314 reference strain, we saw that the most promising peptides were D (10 µM) and AC-1 (10 µM), which showed an important decrease in yeast growth with a significance of *p* < 0.05, as shown in Figure 3.

The growth inhibition induced by these four LL-37 analogue peptides was also significant in all concentrations used (10, 5, 2.5, 1.25, and 0.62 µM) against *C. krusei* ATCC 6558 (Figure 4).

In the same way, when *C. parapsilosis* ATCC 22019 was challenged, the growth inhibition induced by the AC-1 peptide was significant at 10 µM, 5 µM, and 2.5 µM concentrations. The other analogue peptides (AC-2, D, and LL37-1) caused a significant decrease in growth at concentrations of 10, 5, 2.5, 1.25, and 0.62 µM, as shown in Figure 5. It is important to highlight the promising antifungal effect of LL-37 analogue peptides, including azole-resistant strains such as the fluconazole-resistant *C. albicans* 256 and some clinical isolates from women with vulvovaginal candidiasis who presented high MICs to fluconazole (Table 1).

### 3.3. Effect of LL-37 Analogue Peptides in the Biofilm Formation Process

*C. albicans* ATCC 10231 yeasts treated with the LL-37-derived peptides showed a significant reduction in metabolic activity, evidenced by biofilm formation compared to untreated yeasts. At 20, 10, 5, and 2.5 µM concentrations, the four analogue peptides herein studied showed a statistically significant decrease in the biofilm formation (*p* < 0.01). Additionally, at concentrations of 1.25 and 0.62 µM, the metabolic activity declined. Although less evident, it had a significance of *p* < 0.05, as can be observed in Figure 6. Additionally, yeasts treated with different concentrations of amphotericin B used as a control showed a significant decrease in biofilm formation compared to the untreated control group (Figure 6).

### 3.4. Scanning Electron Microscopy (SEM)

*C. albicans* ATCC 10231 yeasts treated with LL-37 analog peptides showed structural alterations, such as cell wall rupture (Figure 7B_1_), bud cell detachment (Figure 7C_1_), and pseudohyphal inhibition. Yeasts not treated with the analogous peptides (control group) presented a more homogeneous structure, and even the formation of pseudohyphae was observed (Figure 7).

## 4. Discussion

In this study, we showed the in vitro antifungal activity of four analogue peptides to human cathelicidin LL-37 against yeasts of the genus *Candida* that reflects a possible therapeutic alternative, favoring advances in the rational design of new peptides as possible therapies to treat superficial and deep mycoses.

The derived peptides herein analyzed (AC-1, AC-2, LL37-1, and D) showed inhibitory activity in vitro against different *Candida* species and even against 20 clinical strains obtained from patients with vulvovaginal candidiasis. The differences observed in the inhibitory processes of each peptide are possibly related to some intrinsic characteristics of the antimicrobial peptides, such as their positive charge (generally +2 to +9) [20] and their cationic nature, which allows these peptides to bind ideally to the anionic charges of cell membranes [21]. On the other hand, the peptides’ hydrophobicity provides them with the ability to insert into microbial membranes causing structural damage. Finally, the amphipathicity or duality presented by these peptides, as they contain apolar and polar regions, results in an increased antimicrobial activity [22,23].

Regarding the synthetic variants of cathelicidin LL-37, the AC-1 peptide showed an outstanding antifungal effect at the five concentrations utilized against *Candida albicans* 256 and at 10 μM for *Candida albicans* SC5314. The AC-1 peptide has an acetylated N-terminal domain in its chemical structure that allows it to insert itself with more affinity in the fungal cell membrane, avoiding an adequate lipid packing and, therefore, increasing its lytic capacity [24].

The AC-2 peptide showed an important antifungal activity against different *Candida* spp. strains, as observed in *C. tropicalis* ATCC 750, *C. krusei* ATCC 6558, *C. parapsilosis* ATCC 22019, and *C. albicans* ATCC 10231. Probably, the action mechanism of AC-2 is different from that of AC-1 since, within its chemical structure, it has one glycine more and it is acetylated and amidated in the carboxyl-terminal position, which confers protection against the microorganism’s proteolytic systems.

Tsai and coworkers (2014) [25] announced the antifungal activity of LL-37 against *C. albicans*, detecting the fungicidal effect from 20 μg/mL (~4.4 μM) to 40 μg/mL (~8.8 μM). Our results corroborate this information since the LL-37 analogue peptides such as LL-37-1 strongly inhibited yeast growth, with MICs observed from 0.07 μM for *C. tropicalis* ATCC 750 to 5 μM for *C. albicans* clinical isolates. It should be noted that when the *Candida albicans* 256 strain was exposed to the LL37-1 peptide, the inhibitory effect was evidenced in the five concentrations analyzed. Possibly, the fact that LL37-1 is amidated in the C-terminal position provides it a protective effect (temporary) and increases its stability against microorganism exonucleases, enhancing its biological activity [26].

Peptide D presented an outstanding antifungal effect, with MICs between 0.15 and 5 µM against different strains of *C. albicans* 256 resistant to azoles, *C. tropicalis* ATCC 750, *C. parapsilosis* ATCC 22019, and some of the clinical isolates. Peptide D has a structural change in the right region of carbon-α (chiral carbon), classifying itself as a positively charged D enantiomer, which allows it to easily bind to the anionic charges of microbial membranes, causing the microorganism’s structural destabilization and promoting the D peptide stability in proteolytic systems, thus enhancing its antimicrobial effect [27].

The present work shows the inhibitory effect of the LL-37 analogue peptides (LL37-1, AC-1, AC-2, and D) in small concentrations (0.62 μM) against strains with a broad resistance profile to azoles such as *C. albicans* 256 and clinical strains from patients with vulvovaginal candidiasis as well as different *Candida* species’ control strains. It should be noted that the purpose of proposing new antimicrobial peptides as possible therapeutic candidates consists of the search for minimum concentrations, in which the risk of cytotoxicity for the host cells could consequently be reduced [25].

Additionally, the peptides analyzed in this work presented an important inhibitory effect on the yeast growth in the planktonic state (range of action from 0.62 µM to 20 µM). Additionally, they strongly decreased the biofilm formation process, which is one of the main virulence factors of different *Candida* species, due to the biofilm limiting the antifungals’ penetration through the extracellular matrix and preventing the host’s proper immune response functioning [28]. Inhibiting biofilm formation is undoubtedly a very important step to control *Candida* infection since the ability to form biofilm leads to a fungal successful persistence that is associated with high mortality rates [29].

AC-1, LL37-1, AC-2, and D peptides showed significant antifungal activity against yeast of the *Candida* genus. It is important to highlight the antifungal effect of these peptides, even in strains resistant to fluconazole, as well as clinical isolates that cause recurrent vulvovaginal candidiasis that have reduced susceptibility to this azole. However, more detailed studies are needed to better understand the interaction of these analogue peptides with pathogenic yeasts. It is crucial to continue with the search for these types of peptides with antifungal properties, optimizing biophysical parameters, maximizing their antimicrobial effect, and minimizing the toxicity to host cells. On the other hand, the analogue peptides may be interesting therapeutic candidates to control infections caused by multiresistant strains to different conventional antifungals.

Finally, scanning electron microscopy (SEM) analysis showed the negative effect of LL37 analogue peptides (AC-1, AC-2, LL37-1, and D) on *C. albicans* yeasts. Important morphological alterations on the cell wall, among others, have been described by some authors studying different antimicrobial peptides against this yeast with promising results [30] Through this methodology it was possible to verify the antifungal effect of the analogous peptides of LL37. However, in the future, it will be necessary to carry out other complementary assays to better understand the site and mechanism of action of peptides on fungal yeasts.

Some research groups have analyzed the LL37 peptide cytotoxicity [31]. However, no data on the specific cytotoxicity of the analogous peptides herein studied have been published so far. Coworkers are currently carrying out studies focused on the possible toxicity of these peptides in human red blood cells and in the fibroblast cell line L929 with promising results after 24, 48, and 72 h after peptide exposure (data in process).

Other formulations could be developed in the future using delivery systems, such as nanoparticles, in order to improve the antifungal effect of the antimicrobial peptides herein studied. Peptides immobilized in nanoparticles are a promising option in fungal infections’ control since this could potentiate the effect of some conventional antimicrobial or antifungal molecules and present a synergistic or additive effect in the infection control [32,33]. It would be interesting to address, in subsequent studies, the possible synergistic or additive effects that the mentioned peptides may present if together they deliver antifungal drugs currently used in clinics. Finally, we suggest that peptides derived from human cathelicidin LL-37 are potential therapeutic candidates due to their rapid mechanism of action and in vitro efficiency in yeast control and are projected as a therapeutic option against candidiasis, a frequent and important mycosis due to the high morbidity and mortality it causes worldwide.

## Figures and Tables

**Figure 1 jof-08-01173-f001:**
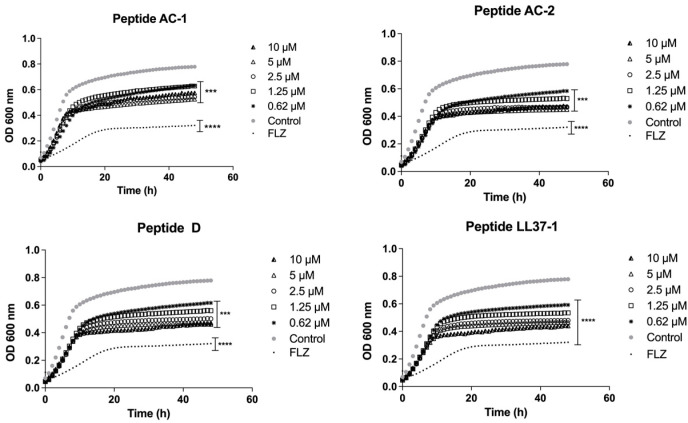
Growth curves of *C. albicans* ATCC 10231. Four LL-37 analogue peptides (AC-1, AC-2, D, and LL37-1) were tested at different concentrations (10, 5, 2.5, 1.25, and 0.62 µM). *C. albicans* ATCC 10231 cells without exposure to antimicrobial peptides were used as a control. Fluconazole (64 µg/mL) was used as antifungal control. Data represent three independent experiments. Statistical significance *** *p* < 0.001; **** *p* < 0.0001 when compared to control.

**Figure 2 jof-08-01173-f002:**
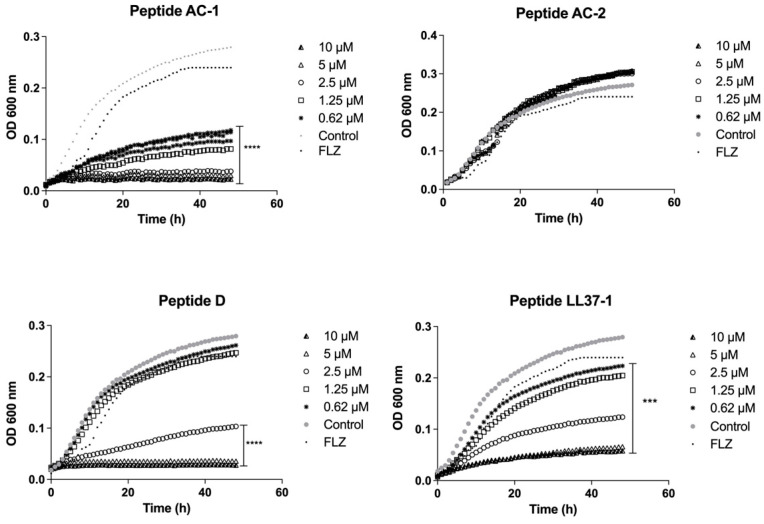
Growth curves of *C. albicans* 256. Four LL-37-derived peptides (AC-1, AC-2, D, and LL37-1) were tested at different concentrations (10, 5, 2.5, 1.25, and 0.62 µM). *C. albicans* 256 cells without exposure to antimicrobial peptides were used as a control. Fluconazole (64µg/mL) was used as antifungal control. Data represent three independent experiments. Statistical significance *** *p* < 0.001, **** *p* < 0.0001 when compared to control.

**Figure 3 jof-08-01173-f003:**
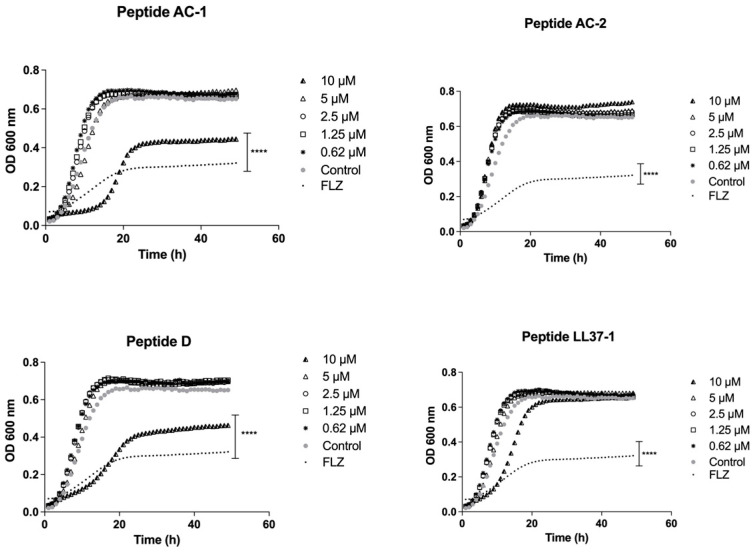
Growth curves of *C. albicans* SC 5314. Four LL-37 analogue peptides (AC-1, AC-2, D, and LL37-1) were tested at different concentrations (10, 5, 2.5, 1.25, and 0.62 µM). *C. albicans* SC 5314 cells without exposure to antimicrobial peptides were used as a control. Fluconazole (64 µg/mL) was used as antifungal control. Data represent three independent experiments. Statistical significance **** *p* < 0.0001 when compared to control.

**Figure 4 jof-08-01173-f004:**
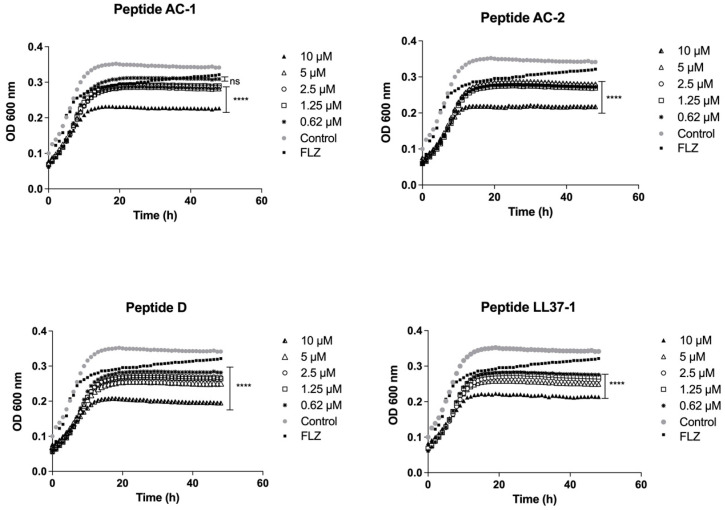
Growth curves of *C. krusei* ATCC 6558. Four LL-37-derived peptides (AC-1, AC-2, D, and LL37-1) were tested at different concentrations (10, 5, 2.5, 1.25, and 0.62 µM). *C. krusei* ATCC 6558 cells without exposure to antimicrobial peptides were used as a control. Fluconazole (64 µg/mL) was used as antifungal control. Data represent three independent experiments. Statistical significance **** *p* < 0.0001 when compared to control.

**Figure 5 jof-08-01173-f005:**
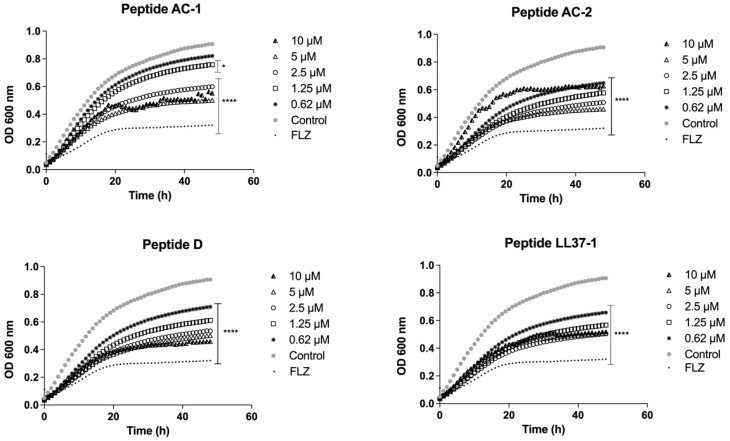
Growth curves of *C. parapsilosis* ATCC 22019. Four LL-37 analogue peptides (AC-1, AC-2, D, and LL37-1) were tested at different concentrations (10, 5, 2.5, 1.25, and 0.62 µM). *C. parapsilosis* ATCC 22019 cells without exposure to antimicrobial peptides were used as a control. Fluconazole (64 µg/mL) was used as antifungal control. Data represent three independent experiments. Statistical significance * *p* < 0.05; **** *p* < 0.0001 compared to control.

**Figure 6 jof-08-01173-f006:**
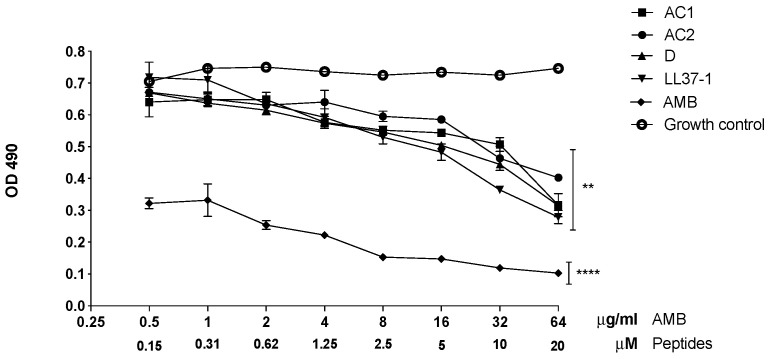
Effect of LL-37 analog peptides on *C. albicans* ATCC 10231 biofilm formation. Colorimetric reaction was read spectrophotometrically at 490 nm. Statistical difference (** *p* < 0.01) was observed in yeasts treated with 20, 10, 5, and 2.5 µM of the derived peptides. At 1.25 and 0.62 µM concentrations, the statistical significance was **** *p* < 0.0001 compared to the growth control (yeasts without exposure to the antimicrobial peptides).

**Figure 7 jof-08-01173-f007:**
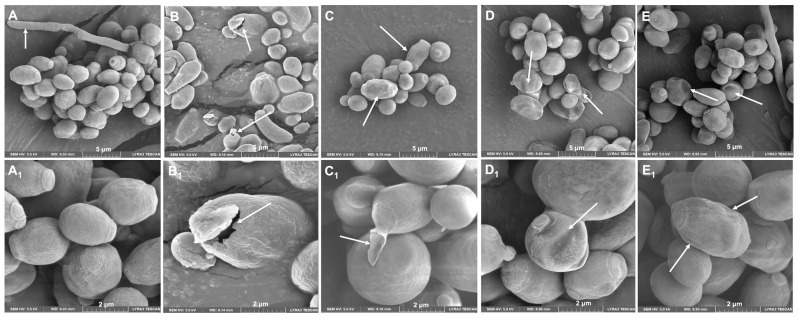
The effect of LL-37 analog peptides (AC-1, AC-2, LL37-1, and D) on *C. albicans* ATCC 10231 structure. SEM micrographs showed that yeasts treated with analog peptides had perturbations on their wall: (**A**) yeast in the absence of peptide; (**B**) treated with analog peptide AC1; (**C**) treated with analog peptide AC2; (**D**) treated with analog peptide LL37-1; and (**E**) treated with analog peptide D. Images (**A**–**E**) were taken at 5 µm magnification; images (**A_1_**–**E_1_**) were taken at 2 µm magnification. White arrows show the damage caused to the yeasts.

**Table 1 jof-08-01173-t001:** Minimum inhibitory concentration (MIC) values of the AC-1-, AC-2-, D-, and LL37-1-derived peptides (expressed in μM). Amphotericin B (AMB) and fluconazole (FLZ) (expressed in μg/mL) were used as antifungal controls. Twenty vulvovaginal candidiasis samples, representing the combination of all clinical isolates in this study.

MIC of LL-37 Analogue Peptides							
Peptide	*C. albicans* ATCC 10231	*C. albicans* SC5314	*C. albicans* 256 (Clinical Strain)	*C. parapsilosis* ATCC 22019	*C. krusei* ATCC 6558	*C. tropicalis* ATCC 750	20 Vulvovaginal Candidiasis Clinical Isolates
AC-1	1.25–0.31 µM	10 µM	5 µM	0.15 µM	0.15 µM	0.15 µM	10–2.5 µM
AC-2	0.31–0.07 µM	5 µM	5 µM	2.5 µM	0.15 µM	0.07 µM	10–2.5 µM
D	2.5–1.25 µM	5 µM	5 µM	5 µM	10 µM	0.15 µM	5–2.5 µM
LL37-1	0.31–0.07 µM	1.25 µM	2.5–5 µM	1.25 µM	1.25 µM	0.07 µM	5–2.5 µM
FLZ	0.25 µg/mL	0.25 µg/mL	8–16 µg/mL	0.5 µg/mL	16 µg/mL	0.5 µg/mL	2–0.5 µg/mL
AMB	0.5–1 µg/mL	0.5–1 µg/mL	4 µg/mL	0.5 µg/mL	0.5 µg/mL	2 µg/mL	2–4 µg/mL

## Data Availability

Not applicable.

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
