# Peer review of "In Vitro Antifungal Activity of LL-37 Analogue Peptides against Candida spp."

_jof, 2022, doi:10.3390/jof8111173_

Round 1
Reviewer 1 Report
Author Pinilla et al. describe "In vitro antifungal activity of LL-37 analogue peptides against Candida spp."
This paper can be accepted after considering the following comments;
1: Line 135, no addition of yeast
2:Line 175 should be biofilm eradication instead of inhibition
3: Table 1 and Figure 2 are not supporting to each other. If the MIC value of the peptide towards C. albicans ATCC10231 is in the range of 2.5 to 0.07 microMolar, then in Figure-1, how does this strain grow in the presence of 10 microMolar (even at 5 microMolar) of each peptide?
4: Similar to the case of other Candida species in Figures -2, 3, 4, and 5.
This is totally wrong beyond the MIC value, how Candida shows growth at the four-fold higher MIC value of peptide.
5: To support Figure 6 author need to do a microscopic (SEM and fluorescence) examination of the biofilm architecture exposed to the peptide.
6: What is the effect of these peptides on the cell morphology of the Candida species?
7: Better to perform viable cell count (e.g. CFU) method instead OD600 measurement of the cell growth.
Author Response
(Reviewer 1)
We appreciate the reviewers´ comments, they are very important to improve the quality of the manuscript. Following, we respond to each concern or suggestion:
1. Line 135, no addition of yeast
Answer: Unfortunately, we do not understand what this concern refers to.
2: Line 175 should be biofilm eradication instead of inhibition
Answer: Change made
3: Table 1 and Figure 2 are not supporting to each other. If the MIC value of the peptide towards C. albicans ATCC10231 is in the range of 2.5 to 0.07 microMolar, then in Figure-1, how does this strain grow in the presence of 10 microMolar (even at 5 microMolar) of each peptide?
4. Similar to the case of other Candida species in Figures -2, 3, 4, and 5.
Answer to questions 3 and 4: Certainly, exists a difference in Candida growth when comparing the data from the two experiments. In the case of the MIC assay (Table 1), performed according to the CLSI guidelines, the antifungal activity is read 24 hours (visual reading) after adding the different concentrations of drugs or molecules tested. In the case of the growth curve (Figure 1), despite using the same concentration (0.5 on the MacFarlan scale) of antifungals and molecules, the conditions change, because in this test yeasts remain in continuous shaking at 100 rpm for 48 hours, a situation that favors the fungal growth. Additionally, the reading is performed at a different optical density (600nm), a situation that may favor or show a greater growth of Candida.
5: To support Figure 6 author need to do a microscopic (SEM and fluorescence) examination of the biofilm architecture exposed to the peptide.
Answer: As part of a new work in progress, we are performing scanning (SEM) and transmission (TEM) microscopy to better understand the mechanism of action of these antimicrobial peptides on the yeast membrane and the biofilm structure of different Candida species.
6: What is the effect of these peptides on the cell morphology of the Candida species?
Answer: Recent tests carried out in our laboratory show that analogous peptides of LL-37 have the capacity to inhibit the formation of pseudohyphae (an important virulence factor) of Candida albicans, corroborating what was described by Wong et al 2011 (PMID: 21889964) who states that through different microscopy techniques was verified the ability of LL-37 and derived peptides to permeabilize fungal membranes.
7: Better to perform viable cell count (e.g. CFU) method instead OD600 measurement of the cell growth.
Answer: We agree with the reviewer's comment, we will take it into account for future trials, thanks.
Reviewer 2 Report
The manuscript "In vitro antifungal activity of LL-37 analogue peptides against Candida spp." shows the direct antifungal effects of four antimicrobial peptides against various Candida species, which are of clinical relevance.
Given the rise of wide-spread antibiotic resistance amongst bacteria and fungi, finding new antibiotics is of great interest. The efficacy of their peptides within RPMI media is noteworthy, as many AMPs become inactivated in nutrient rich cell culture media.
There are some language issues in the paper (mostly typos, such as D.O. rather than OD) which can be fixed. The discussion is well written.
While the study is interest, this Reviewer would recommend either adding some more experiments, or adjusting the text to clarify some questions.
Major comments:
The authors give a thorough description of the description of the structures of their peptides. However, for this Reviewer, it would be easier to have a figures displaying the molecular structure of each peptide would make it easier to understand the differences between each compound.
Figures 1-5 are quite similar. The difference between each figure is of course the species/type stain but for the reader it makes it unfortunately rather boring and a bit tedious to go through.
I would recommend that the authors perhaps merge figures 1-5 into a single figure and keep the data shown in Figures 1-5 in the supplement instead. For example, I would choose a single concentration of each peptide (such as 10µM) and plot the growth curve of each species in the same graph (see the attached PDF for an example of what I mean).
The authors look at the possible antifungal effects of their LL-37 analogues, but do not make any comments as to whether or not they have improved upon the anti-candida ability of LL-37. Nor do they comment on possible improved resistance to bacterial proteases. This could be another avenue for the authors to consider looking into.
Minor Comments:
Line 19: The phrasing "an international problem; especially, Candida" is odd and can be rewritten for clarity/grammar
Line 47: The reference to 2017 appears to be incorrect as the article is from 2018. Reference number 19 also has the incorrect publishing year. Please go through references to check for typos.
Line 67: LL-37 is the only human cathelicidin, so to reference it as being from the "human cathelicidin family" is odd.
Line 69: There appears to be a hyphen in the middle of the amino acid sequence. Why?
Line 98: the authors mention degradation of peptides by proteases released by microbes, but host AMPs can also be inactivated by host proteases
Lines 114-126: This is likely a formating issue as the text is all in italics. However, information on the cytotoxicity of these peptides towards human cells is missing.
Lines 122-126: This information regarding the peptide structure is already given in lines 99-103
Line 126: Here it is written that peptide D is 21 amino acids long, but in Line 103, it's written that it's 25 amino acids long. Which is correct?
Line 129: I cannot find a reference for C. albicans strain 256? Is this a type strain or clinical isolate?
Lines 130-131: please describe how the clinical isolates of C. albicans were identified as C. albicans
Line 135: the word "shacked" is incorrect. It should be "shaken"
Lines 146-147: I find the description of how the McFarland density of CFUs difficult to follow. From the phrasing it seems like you made a McFarland density of 0.08-0.1 of your inoculum and claim this is 1x108 CFU/mL. But that is not correct. A 0.5 McFarland is 1x108 CFU/mL. Please rephrase this section to make it more clear for the reader.
Line 149: Please change "y" to "and".
Lines 161-162: See previous comment on McFarland standard
Lines 199-205: This information is shown in Table 1, I would not repeat the data twice.
Figures 1-5: the label on the Y-axis is incorrect (not D.O., but OD). In addition, in the figure legends, it says that "data represent three independent experiments", but it's not specified whether each point is the mean or the median. It would also be nice to show the error bars for each point, but I would understand if it makes the graph impossible to read.
Figure 2 and 4: Interestingly, your Candida isolates don't grow as well as compared with the other type strains (OD is half of that as the other strains). Is this characteristic of each strain? Or might it have something to do with fluconazole resistance?
Figure 6: The "growth control" group is confusing and shouldn't be placed where it is. Rather the x-axis should start at 0, and the values for 0 should be taken from the "growth control" group.
The x-axis also needs to have better labelling to show that the concentrations are for either the peptides, or for amphotericin B.
The y-axis labelling needs to be changed, so that OD is written correctly, and the correct wavelength is written. At the moment, the figure legend and the y-axis state different wavelengths (492nm vs. 490nm).
Lines 313-317: This experiment can be quickly carried out and would improve this manuscript. If possible, I would recommend that the authors perform this, as it would be quick and strengthen the manuscript.
Lines 344-346: The authors observed a significant reduction in metabolism with biofilm treated with their peptides. However, I am curious as to whether they also observed a significant reduction in biofilm volume? For example if by the crystal violet assay, whether they would see less bound CV when treating pre-formed biofilms with their AMPs. This would also strengthen the manuscript, as the authors could show that not only does their peptide induce lower metabolism in biofilm, but also can dissolve existing biofilm. This would make the peptides more interesting candidates for treating candidiasis.
Line 369: Most AMPs work within minutes after application. So stating that they work within 24 hours is a bit excessive
Author Response
We appreciate the reviewer comments, they are very important to improve the quality of the manuscript. Following, we respond to each concern or suggestion:
There are some language issues in the paper (mostly typos, such as D.O. rather than OD) which can be fixed. The discussion is well written.
Answer: Throughout the manuscript the term D.O. was replaced by OD, as suggested by the reviewer.
2. The authors look at the possible antifungal effects of their LL-37 analogues, but do not make any comments as to whether or not they have improved upon the anti-candida ability of LL-37. Nor do they comment on possible improved resistance to bacterial proteases. This could be another avenue for the authors to consider looking into.
Answer: From lines 69 to 89 of the introduction, the antimicrobial properties of the “original” LL-37 peptide, as well as its origin and production within the body, are clearly and extensively described. Additionally, in the following lines the use of analogous peptides instead of the original peptide is justified. As its name indicates, LL-37 peptide has a structure of 37 amino acids that makes it an easy target for proteases secreted by some microorganisms as a defense mechanism. In the case of smaller analog peptides, it is easier to circumvent this resistance mechanism, as explained in lines 98 and 108 of the introduction.
3. Line 19: The phrasing "an international problem; especially, Candida" is odd and can be rewritten for clarity/grammar
Answer: Text corrected following reviewer's recommendations
4. Line 47: The reference to 2017 appears to be incorrect as the article is from 2018. Reference number 19 also has the incorrect publishing year. Please go through references to check for typos.
Answer: In the case of the reference of line 47, we clarify that the study was carried out in 2017, although the manuscript was published until 2018, as we express in our manuscript. On the other hand, the year of publication of reference number 19 was corrected as well noted by the reviewer.
5. Line 67: LL-37 is the only human cathelicidin, so to reference it as being from the "human cathelicidin family" is odd.
Answer: As suggested by the reviewer, the term "cathelicidin family" was corrected throughout the text.
6. Line 69: There appears to be a hyphen in the middle of the amino acid sequence. Why?
Answer: Certainly your assessment is correct, in the original document the sequence had been divided into two lines and the editing program placed a hyphen automatically, now in the journal format the sequence remained on one line and the hyphen was removed.
7. Line 98: the authors mention degradation of peptides by proteases released by microbes, but host AMPs can also be inactivated by host proteases.
Answer: Important observation from the reviewer, actually host proteases can also affect antimicrobial peptides, to better understand, additional information was added in the text on line 98.
8. Lines 114-126: This is likely a formatting issue as the text is all in italics. However, information on the cytotoxicity of these peptides towards human cells is missing.
Answer: It was indeed an inadvertent formatting error. The information on the peptides cytotoxicity was added in the manuscript discussion in lines 381-385.
9. Lines 122-126: This information regarding the peptide structure is already given in lines 99-103
Answer: Agreeing, this information is in lines 99 and 103. However, in lines 122 and 126, in addition to express the number of amino acids that the peptide has, its sequence is also shown.
10. Line 126: Here it is written that peptide D is 21 amino acids long, but in Line 103, it's written that it's 25 amino acids long. Which is correct?
Answer: Peptide D has 25 amino acids; this information was corrected in line 128.
11. Line 129: I cannot find a reference for albicans strain 256? Is this a type strain or clinical isolate?
Answer: Yes, it is a clinical strain with reduced susceptibility to fluconazole.
12. Lines 130-131: please describe how the clinical isolates of C. albicans were identified as C. albicans
Answer: The required information on the identification of C. albicans was added to the text on lines 136-137.
13. Line 135: the word "shacked" is incorrect. It should be "shaken"
Answer: Thanks for your observation, the word was corrected in the text.
14. Lines 146-147: I find the description of how the McFarland density of CFUs difficult to follow. From the phrasing it seems like you made a McFarland density of 0.08-0.1 of your inoculum and claim this is 1x108 CFU/mL. But that is not correct. A 0.5 McFarland is 1x108 CFU/mL. Please rephrase this section to make it more clear for the reader.
Answer: The manuscript was modified in line 154.
15. Line 149: Please change "y" to "and".
Answer: Correction was made in the text.
16. Lines 161-162: See previous comment on McFarland standard
Answer: Corrected
17. Lines 199-205: This information is shown in Table 1, I would not repeat the data twice.
Answer: The information placed in these lines explains the results obtained in the assay for a better understanding of the reader.
18. Figures 1-5: the label on the Y-axis is incorrect (not D.O., but OD). In addition, in the figure legends, it says that "data represent three independent experiments", but it's not specified whether each point is the mean or the median. It would also be nice to show the error bars for each point, but I would understand if it makes the graph impossible to read.
Answer: The label on the Y-axis was corrected, the plotted points are the mean of the experiments performed.
19. Figure 2 and 4: Interestingly, your Candida isolates don't grow as well as compared with the other type strains (OD is half of that as the other strains). Is this characteristic of each strain? Or might it have something to do with fluconazole resistance?
Answer: Yes, according to the reviewer's comment, there is variability in the growth of the strains, where the growth of the ATCC Candida albicans strains (ATCC 10231 and SC5314) stands out. However, the growth of the C. albicans 256 isolate was much lower compared to other C. albicans strains, even lower than other Candida species. As the reviewer says, possibly due to its resistance to fluconazole or some other characteristic of this strain.
20. Figure 6: The "growth control" group is confusing and shouldn't be placed where it is. Rather the x-axis should start at 0, and the values for 0 should be taken from the "growth control" group.
The x-axis also needs to have better labelling to show that the concentrations are for either the peptides, or for amphotericin B.
Answer: Figure 6 corrected, reviewer's instructions followed.
21. The y-axis labelling needs to be changed, so that OD is written correctly, and the correct wavelength is written. At the moment, the figure legend and the y-axis state different wavelengths (492nm vs. 490nm).
Answer: OD and wavelength were corrected on the graph following the protocol established by Pierce, et al., 2009.
22. Lines 313-317: This experiment can be quickly carried out and would improve this manuscript. If possible, I would recommend that the authors perform this, as it would be quick and strengthen the manuscript.
Answer: It is not clear which experiment the reviewer recommends.
23. Lines 344-346: The authors observed a significant reduction in metabolism with biofilm treated with their peptides. However, I am curious as to whether they also observed a significant reduction in biofilm volume? For example, if by the crystal violet assay, whether they would see less bound CV when treating pre-formed biofilms with their AMPs. This would also strengthen the manuscript, as the authors could show that not only does their peptide induce lower metabolism in biofilm, but also can dissolve existing biofilm. This would make the peptides more interesting candidates for treating candidiasis.
Answer: Interesting suggestion from the reviewer. We may include these analyzes in future studies to show whether antimicrobial peptides, in addition to decreasing biofilm metabolism, also decrease or dissolve existing biofilm. If the editor considers important to add this essay, please let us know.
24. Line 369: Most AMPs work within minutes after application. So stating that they work within 24 hours is a bit excessive.
Answer: this statement was removed from the manuscript (less than 24 hours)
Round 2
Reviewer 1 Report
Required more experimental data to support this study
Author Response
Dear reviewer 1,
We appreciate your comments and suggestions to improve our work. However, for reasons of financial resources and limited time, we could only perform scanning microscopy (SEM) which, according to the facility core of our institution, could be ready in 2 to 3 weeks. Would this proposal be good for you?
Kind regards,
Reviewer 2 Report
The authors of the paper "In vitro antifungal activity of LL-37 analogue peptides against Candida spp." have taken into consideration the critiques and suggestions given by this Reviewer.
However, before accepting, I would highly recommend that they fix some minor issues before publishing
Perhaps something was wrong in the PDF version which I received, but I did not see any changes made to figures 1-5, where the authors stated that they changed the labelling on the y-axis to fix the spelling of OD (still are labelled as D.O.). However, in the version that I have, this has not been corrected. I would change this to avoid having to publish an erratum if it is published as it is.
Author Response
Dear Reviewer 2,
The figures were modified in the manuscript following your recommendations. Since the first revision, the figures had been updated individually in the attached documents, but due to an involuntary error they were not updated in the manuscript as such, we hope that this time you will be able to see the suggested changes.
Thanks for your patience and kindness.
Best regards,
Julian E. Muñoz
Round 3
Reviewer 1 Report
The author agrees to perform the SEM analysis in the presence and absence of peptides. Let them finish and updates the finding in the paper.
Author Response
Dear reviewer,
According to what was requested by you, the scanning microscopy (SEM) assay was performed and included in the document.
Thank you for helping us improve our manuscript.
kind regards,
Round 4
Reviewer 1 Report
Congratulations for addressing all comments